# What Do We Know about Early Management of Sepsis and Septic Shock in Polish Hospitals? A Questionnaire Study

**DOI:** 10.3390/healthcare9020140

**Published:** 2021-02-01

**Authors:** Łukasz J. Krzych, Agnieszka Wiórek, Paweł Zatorski, Karol Gruca, Karina Stefańska-Wronka, Janusz Trzebicki

**Affiliations:** 1Department of Anaesthesiology and Intensive Care, Faculty of Medical Sciences in Katowice, Medical University of Silesia, 14 Medyków Street, 40-752 Katowice, Poland; 2First Department of Anaesthesiology and Intensive Care, Medical University of Warsaw, 02-005 Warsaw, Poland; zator22@wp.pl (P.Z.); janusz.trzebicki@wum.edu.pl (J.T.); 3Students’ Scientific Society of the Department of Anaesthesiology and Intensive Care, Faculty of Medical Sciences in Katowice, Medical University of Silesia, 14 Medyków Street, 40-752 Katowice, Poland; gruca.karol61@gmail.com; 4Department of Anesthesiology and Intensive Therapy, District Hospital in Poznań, 60-479 Poznań, Poland; karina_wronka@poczta.onet.pl

**Keywords:** guidelines compliance comparison, intensive care unit, sepsis guidelines adherence, sepsis and septic shock management, Surviving Sepsis Campaign

## Abstract

Background: Sepsis and septic shock are medical emergencies with a high risk of poor prognosis. We investigate the correspondence between Surviving Sepsis Campaign (SSC) guidelines and clinical practice in Poland, with special attention given to differences between ICU and non-ICU environments as well as regional variations within the country. Methods: A web-based questionnaire study was performed on a random sample of 60 hospitals from the three most populated regions in Poland—Masovia, Silesia, and Greater Poland. A 19-item questionnaire was built based on the most recent edition of SSC guidelines. Results: Sepsis diagnosis was primarily based on clinical evaluation (ICUs: 94%, non-ICUs: 62%; *p* = 0.02). There were significant differences between ICUs and non-ICUs regarding taking blood cultures for pathogen identification (2-times more frequent in ICUs) and having hospital-based operating procedures to adjust antimicrobial treatment to a clinical scenario (a difference of 17%). Modification of empiric antimicrobial treatment was required post-ICU admission in 70% of cases. ICUs differed from non-ICUs with regard to the methods of fluid responsiveness assessment and the types of catecholamines and fluids used to treat septic shock. The mean fluid load applied before the implementation of catecholamines was 25.8 ± 10.6 mL/kg. Norepinephrine was the first-line agent used to treat shock, and balanced crystalloids were preferred in both ICUs and non-ICUs. Conclusion: Compliance with SCC guidelines in Polish hospitals is insufficient, especially outside ICUs. There is a need for education among healthcare professionals to reach at least an acceptable level of knowledge and attitude in this field.

## 1. Introduction

Sepsis and septic shock are medical emergencies with a high risk of poor prognosis. Mortality in septic shock reaches 50% [1] and remains at this level with the passing years [2,3]. Due to the progress made in understanding the pathophysiology of sepsis, the definition of sepsis and septic shock was revised in 2016 as part of the Sepsis-3 initiative [4]. In turn, as a part of the Surviving Sepsis Campaign (SSC), updated guidelines were published on optimal diagnostic and therapeutic management [5]. These international recommendations describe the so-called “care bundles” (CB), comprising procedures to be performed in case of suspected or confirmed sepsis in the first, third and sixth hours after their identification [5]. CBs usually need to be adjusted to local needs and possibilities and may differ between intensive care units (ICUs) and other hospital wards due to differences in equipment, personnel, and procedures. The “Hour-1 Bundle” (H1B) describes the initial steps to be taken in a “golden hour” when sepsis is suspected and includes early identification, collecting blood for microbiological cultures, prompt administration of broad-spectrum antimicrobial agents, and complex personalized hemodynamic management [6].

Rapid implementation of CBs is beneficial to the patient [7,8,9] and may account for even a 1/3 decrease in mortality from septic shock [10]. However, this can only be achieved with good adherence and compliance with the recommendations [11].

In this study, we attempt to investigate the correspondence between the current guidelines for sepsis and septic shock management and actual clinical practice in random hospitals in Poland, with special attention given to differences between ICU and non-ICU environments as well as regional variations within the country.

## 2. Materials and Methods

We performed a web-based questionnaire study under the auspices of the section of Intensive Care Medicine of the Polish Society of Anaesthesiology and Intensive Therapy. The study group comprised ICU directors for adults from the three most populated regions in Poland, i.e., Masovia, Silesia, and Greater Poland (Figure 1) [12]. The invitation for participation, with an interactive link to the questionnaire, was sent twice by e-mail between March and August 2020. After a failed second attempt in e-based communication, an additional phone call was performed by the investigators in each region to renew the invitation and to remind the ICU directors of the study procedures. The final response rate was 45.5% (i.e., 60/132). Only one response regarding center-specific procedures given by the ICU director of each hospital was recorded.

The 19-item questionnaire was constructed by two investigators to evaluate the level of compliance with recommendations for sepsis/septic shock management at hospital level in ICU and non-ICU settings. The questions were built based on the most recent edition of SSC guidelines to ensure the high quality of the data and their agreement with evidence-based medicine data [5].

The study was voluntary and anonymous. Under sections 21 and 22 of the Act of 5 December 1996 on the Medical Profession, due to the noninterventional design of the study, no approval of the Ethics Committee was required [13].

Statistical data were recorded using licensed MedCalc version 17.2 (MedCalc Software bvba, Ostend, Belgium) statistical software. Qualitative variables were described with frequencies and percentages. Between-group differences for categorical variables were assessed using the chi-squared test. A *p*-value of <0.05 was considered statistically significant.

## 3. Results

We received 60 questionnaires from hospitals representing the Masovia (42%), Greater Poland (33%), and Silesia (25%) regions (Figure 1).

Basic data regarding participating units are depicted in Table 1.

Table 2 presents the methods used for sepsis screening in participating hospitals. We found statistically significant differences in using clinical evaluation and the implementation of the National Early Warning Score 2 (NEWS2) between ICUs and non-ICUs: clinical evaluation was applied in 94% of ICUs and only in 62% of non-ICUs; infrequent application of NEWS2 was related to lack of rapid response teams (RRTs), which were available only in 17% of hospitals (Figure 2).

The strategies related to antimicrobial treatment are shown in Table 3. We found significant differences between ICUs and non-ICUs regarding taking blood cultures for pathogen identification (2-times more frequent in ICUs), having hospital-based operating procedures to adjust antimicrobial treatment to a clinical scenario (a difference of 17%), and using probiotics (i.e., selected strains of live microorganisms that, when consumed, are beneficial to health through the regulation of the immune system), prebiotics (i.e., nondigestible components that act as stimulants for the growth and activity of advantageous microorganisms) [15], and antibiotics (more frequent in ICUs). More to the point, in approximately 70% of cases, empiric antimicrobial treatment was modified after ICU admission, mainly due to lack of therapeutic effects and the wider therapeutic options available in the ICU (Table 2).

Table 3 and Table 4 present the hemodynamic management procedures for sepsis and septic shock. Figure 3 shows the frequency of application of methods of fluid responsiveness assessment that are specific to the ICU environment (Figure 3). We found that ICUs differed significantly from non-ICUs with regard to the methods of fluid responsiveness assessment and types of fluids used for volume expansion (Table 3), types of catecholamines and vasopressors used to treat septic shock (Table 4), application of the Vitamin C, Thiamine and Steroids in Sepsis (VICTAS) protocol (Figure 4), and the use of extracorporeal blood purification techniques (Figure 5). Hospital-based fluid therapy algorithms were available in only 13% of non-ICUs. Clinical state and arterial blood pressure were the most frequently used methods to evaluate fluid responsiveness. Dynamic techniques were infrequently applied in the ICUs. The mean fluid load applied before the implementation of catecholamines was 25.8 ± 10.6 mL/kg within the first 3 h. Noteworthy, norepinephrine was the first-line agent used to treat shock, and balanced crystalloids were preferred both in ICU and non-ICU settings. Extracorporeal blood purification techniques were unpopular adjuncts to hemodynamic support.

Investigating the variations in sepsis management within ICUs and non-ICUs between regions, we revealed that there were no statistically significant differences between Greater Poland, Silesia, and Masovia. However, there were a few differences between ICUs and non-ICUs within the regions that concerned the application of the NEWS2 score as the sepsis screening method, use of catecholamines in sepsis and septic shock management, monitoring of inflammatory markers, taking blood cultures, and protocolization of antimicrobial treatment (Appendix A).

## 4. Discussion

This questionnaire study aimed to assess the correspondence between the current guidelines for sepsis and septic shock management in their early phase and actual clinical practice in a random sample of hospitals from the three most populated regions in Poland. We found that there was only a fair level of overall compliance in ICU settings and a rather poor level of overall compliance in non-ICUs; these observations were unrelated to the geographical location of the hospitals.

As prevention is always better than cure, a vital part of successful sepsis management is limiting nosocomial transmissions of microorganisms between patients with proper personal hygiene and equipment disinfection; that aspect has long been an issue of great concern [16]. Since the introduction of the newest 2016 edition and their 2018 update, several analyses have been published regarding compliance with Surviving Sepsis Campaign guidelines [17]. SSC sepsis CBs have been developed to simplify the intricate and time-consuming process of translating single published reports, congress presentations, and lectures into universally, internationally approved recommendations and then into clinical practice in hospitals around the world. Bundle adherence has already proven to improve sepsis survival and cut therapy costs [18]. The reduced mortality, cost savings, and improved hospital and ICU lengths of stay have been seen across developed countries like the United Kingdom and Spain and also in developing countries like India, Brazil, and China [18]. Unfortunately, that does not mean that overall bundle compliance is particularly high. Noncompliance is mainly related to delays in sepsis recognition, which leads to delays in treatment application and, therefore, missing the treatment timeframes. This makes it difficult to achieve an optimal effect, as most sepsis care bundles are constructed such that the omission or delay of any element makes the rest of the bundle elements less likely to provide recovery. Similar observations are made in our study.

Following the H1B protocol step by step, immediate attention is targeted at infection control. The selection of a set of antibacterial drugs in combination therapy may accelerate the eradication of pathogens and the endotoxins they produce [5]. At this stage of the procedure, it is usually necessary to initiate the therapy outside the ICU, based solely on the patient’s clinical picture and other simple criteria such as the (quick) Sequential Organ Failure Assessment Score (qSOFA) score or NEWS2 [19]. These steps should be taken by rapid response teams that aim for early identification and prompt management of emergencies outside the ICU [20]. Unfortunately, in our study, only 17% of hospitals had RRTs, and screening with NEWS2 was never applied in 81% of non-ICUs; for qSOFA, it was 44%. Studies have shown a relationship between the early initiation of antibiotic therapy (within the first hour of developing hypotension) and a decrease in mortality in shock patients; studies have also shown an increase in mortality with each hour of delay in the initiation of empirical, broad-spectrum antibiotic therapy [21,22,23]. The introduction of targeted antibiotics is rarely possible on sepsis identification; that is why collecting blood for microbiological cultures is vital in setting the subsequent therapeutic path. In our study, we observed very good tendencies for the frequent collection of blood culture samples. However, one ought to remember that antimicrobial treatment in sepsis has various aspects. The initial treatment should be broad enough to cover the most prevalent organisms for the septic episode during the initial resuscitation. Following blood culture results, de-escalation and narrowing of the therapeutics should be considered. The question regarding therapeutic effects appears more related to the later phases of source control and not the initial resuscitation.

Damage to the glycocalyx, uncontrolled fluid shifts between compartments, and vasoplegia can generate hypovolemia. It causes hemodynamic changes that require an intensive supply of balanced crystalloids to maintain the circulation volume [24]. If the mean arterial pressure (MAP) is <65 mmHg and the concentration of lactate exceeds 4mmol/L, it is necessary to implement liberal crystalloid fluid therapy at the dose recommended in SSC guidelines (i.e., 30 mL/kg). In addition, fluid losses should be replenished with a supply of albumin, which is a colloid recommended for intravascular volume replenishment, due to reports suggesting a reduction in mortality among patients who were given albumin within 6 h after the diagnosis of septic shock [25]. Hypotension persisting despite implemented fluid therapy requires the use of vasopressors, for example norepinephrine, and argipressin to reach a perfusion pressure target of MAP of at least 65 mmHg. In our study, we found that norepinephrine and balanced crystalloids were frequently used as first-line agents. Additionally, the fluid volume transfused was 25 mL/kg, which should be considered adequate for initial resuscitation. Further treatment should be tailored to the patient’s needs after an assessment of fluid responsiveness. To reach this goal, several dynamic methods have been suggested in the literature [26]. Unfortunately, in our study, the adherence to current recommendations regarding hemodynamic monitoring was rather poor. It should also be underlined that there still are discrepancies in terms of utilizing unbalanced crystalloids for volume resuscitation. The issue of optimal fluid therapy (especially in the application of crystalloid fluid boluses) has been considered challenging in the past. A report by Rivers et al. suggested that early goal-directed therapy (EGDT) is superior in terms of short- and long-term outcomes in patients with severe sepsis and septic shock, who, in this study protocol, received significantly more fluids within the initial six hours of the diagnosis [27]. However, in recent years, randomized clinical trials have been conducted worldwide in the form of the Protocolized Care for Early Septic Shock (ProCESS) trial in the US, the Australasian Resuscitation in Sepsis Evaluation (ARISE) study, the Protocolised Management in Sepsis (ProMISE) trial in the UK, and the Fluids and Catheters Treatment Trial (FACTT) study, which have revealed that in the general population of patients with severe sepsis and septic shock, early goal-directed therapy was not beneficial in terms of the outcome when compared with the usual resuscitation that includes a standardized and accepted amount of fluids to be delivered, leading to the revision of the universally applicable sepsis care bundles by the Surviving Sepsis Campaign committee (Appendix A) [28]. The FACTT study was centered around patients with acute lung injury that was also associated with sepsis and septic shock. It reported that the use of a conservative fluid-management protocol aimed at lower central venous pressure or a pulmonary–artery occlusion pressure target resulted in improved lung function and a shortened duration of mechanical ventilation and intensive care without an increase in adverse events, as compared with liberal fluid supplementation targeting higher intravascular filling pressures [29]. However, further research is still required. In our study, the lack of universal hospital-based fluid therapy algorithms may result from the fact that extensive fluid resuscitation and further therapy usually takes place within the ICU as only the ICU environment allows for the optimal methods of fluid responsiveness assessment and fluid therapy monitoring that thoroughly follows the best accessible algorithm provided by the Surviving Sepsis Campaign. Should the need appear for fluid therapy in any other medical or surgical ward before probable admission to the ICU for further treatment, the ROSE (resuscitation, optimization, stabilization, and evacuation) protocol is the implemented strategy of choice [30].

Searching within the reports published within 5 years from the 2016 SSC guidelines, we came across numerous studies describing efforts to introduce SSC guidelines into daily practice. In one of the latest studies on this experience, Igiebor et al. [31] described the impact of the Sepsis Intervention Protocol (SIP) when introduced to emergency departments (EDs). The goal of SIP is to increase the adherence to 3-h and 6-h CBs. They compared the period of 14 months before SIP introduction to a time frame of 11 months after SIP introduction and noticed a statistically significant drop in sepsis mortality, from 40% to 29%. Particular elements that are meant to accelerate response time are worth implementing as widely as possible, such as the “ED sepsis kit”. It contains 2 L of crystalloids and a timer to keep up the time-sensitive therapeutic interventions like lactate concentration measurements. It also includes a checklist to be completed by the nurses and physicians for real-time feedback, leading to better adherence.

A slightly earlier report also focused on the ED practice that was published in 2017 by Moghaddam et al. The practice focuses on a standardized checklist based on SSC protocols, with items categorized into diagnostic and treatment measures. Aspects include checking the vitals within 20 min of ED admission; measuring glycemia, arterial blood gas (ABG) parameters, and urine output; inserting a central venous line, with central venous pressure (CVP) checks in the first 2 h of admission; blood culture testing; high flow oxygen; fluid therapy; broad-spectrum antibiotics; intravenous vasopressor infusion. Emergency medicine residents were first evaluated by their compliance with the protocol; then, they were trained during workshops on their shortcomings and re-evaluated. The results of the re-evaluation showed improved compliance, shorter time from admission to diagnosis, and increased mean knowledge scores. What started as a “fair” to “poor” adherence improved into “good” and “excellent” in multiple items included within the evaluated sepsis checklist [32]. This type of uniform training should be recommended to Polish hospitals to improve their overall performance in terms of sepsis management.

A recently published randomized clinical trial (VICTAS) aimed to determine if the combination of vitamin C, hydrocortisone, and thiamine improves the prognosis and outcome in patients with septic shock compared with hydrocortisone alone [33]. The combination of high-dose intravenous vitamin C, thiamine, and hydrocortisone was popularized by a single-center retrospective before-and-after study performed on 94 patients with severe sepsis or septic shock, published in 2017 [34]. This intervention was associated with an increased number of vasopressor-free days and decreased in-hospital mortality [34]. However, despite repeated testing by numerous finalized or ongoing studies, the protective effect of this drug combination has not yet been decisively confirmed [33,35,36,37]. Even so, “Marik’s protocol” is frequently implemented among our respondents, particularly in the ICU setting.

One ought to bear in mind that sepsis and septic shock are multidisciplinary challenges, and healthcare workers of different areas of expertise should always be up-to-date with the newest developments in the field. In countries with human resource shortages, frontline staff may be formed by senior medical students, interns, or nonphysician clinical assistants so an adequate education should be the focus from the very beginning of medical training [38]. A delay in recognition of sepsis symptoms while the patient is examined by the first-contact medical team may lead to irreversible deterioration of the overall outcome prognosis. The study published by MacMillian et al. described the development of a hospital-wide automated sepsis alert system. It was implemented to improve compliance with sepsis guidelines, especially among ED staff, critical care nurses, internal medicine physicians, and intensivists [39]. The patients included in the study were monitored with electronic surveillance, automated serial assessments of white blood cell, platelet counts, serum creatinine, coagulation parameters, and lactate levels, and were assessed by an assigned bedside nurse who recorded the values of blood pressure, temperature, respiratory rate, PaCO_2_, and urine output. In the case of suspected sepsis, the sepsis response team was called and was expected to arrive at the patient’s bedside within 15 min to evaluate the patient and implement sepsis care-bundle procedures. Although the cited study did not record a statistically significant change in sepsis outcome or ICU length of stay, the participants reported workflow improvement and increased levels of confidence when dealing with a septic patient [39]. These actions should be recommended to our hospitals and should cover the preparation of hospital-based standard operating procedures and clear and structured algorithms and protocols, enabling goal-directed management of sepsis and septic shock. We confirm that there is still room for improvement, especially in non-ICUs.

### Study Limitations

This study has a few limitations. First of all, it has a limited number of participants, which may not be representative of the entire population of Poland. We tried to reduce this shortcoming by sending our electronic invitation twice by e-mail; additional phone calls were also performed. We also focused on the three most populated regions in Poland to minimize this bias. Secondly, as in every questionnaire study, respondents may answer in a preconceived manner. The questionnaire was self-filled, so we cannot entirely exclude the effect of subjectivism on the given answers. Finally, only anesthesiologists answered the questions regarding procedures outside the ICU, but we believe that they were aware of how sepsis is managed in their hospitals.

## 5. Conclusions

Compliance with international guidelines on sepsis diagnostics and treatment in a random sample of Polish hospitals is insufficient, especially outside ICUs. There is an urgent need for education among healthcare professionals to reach at least an acceptable level of knowledge and attitude in this field. There is room for improvement in sepsis and septic shock management at its early phase.

## Figures and Tables

**Figure 1 healthcare-09-00140-f001:**
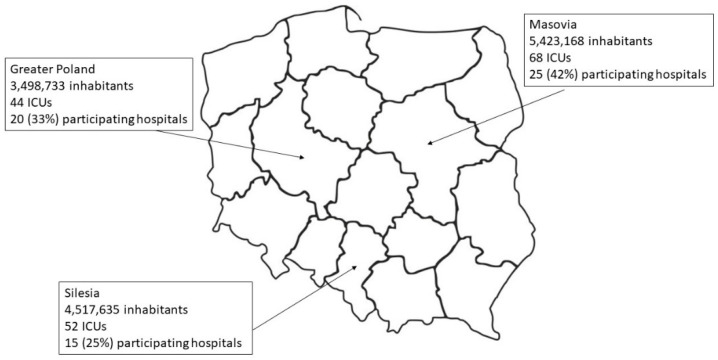
Study group origin and demographics.

**Figure 2 healthcare-09-00140-f002:**
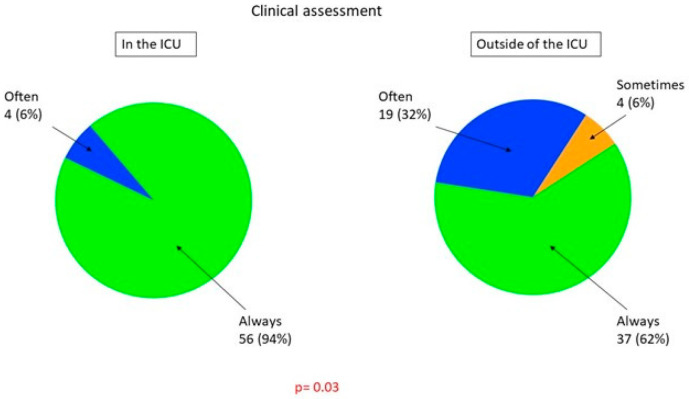
Methods of sepsis screening used in the ICU and outside the ICU. The Quick Sequential Organ Failure Assessment Score (qSOFA) includes one point for each of the following: respiratory rate ≥ 22, SBP ≤ 100 mm Hg, and altered mental status. For screening purposes, a cut-off of two points is used [14]. National Early Warning Score 2 (NEWS2) assigns points to measurements of respiratory rate, SpO_2_, air or oxygen ventilation, systolic blood pressure, pulse, state of consciousness, and body temperature, with increasing severity the higher the calculated sum [14]. Inflammatory parameters include any of the following: C-reactive protein (CRP), procalcitonin (PCT), interleukin 6 (Il-6), tumor necrosis factor (TNF), white blood cell count (WBC), and the neutrophil–lymphocyte ratio (NLR).

**Figure 3 healthcare-09-00140-f003:**
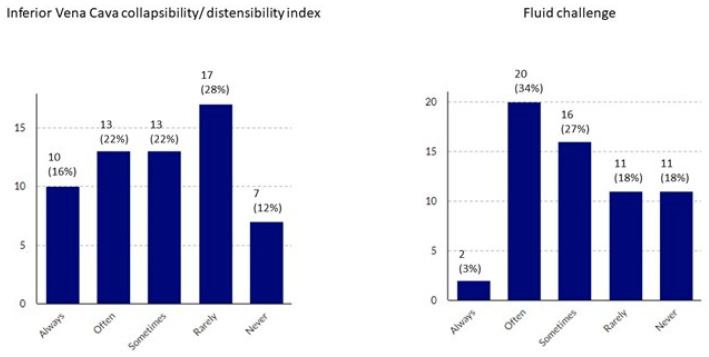
Methods of fluid responsiveness assessment utilized in the ICU environment.

**Figure 4 healthcare-09-00140-f004:**
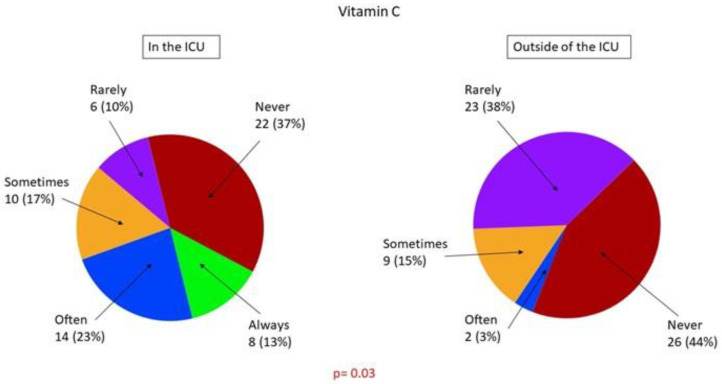
Application of the Vitamin C, Thiamine and Steroids in Sepsis (VICTAS) protocol in the ICU and outside the ICU environment.

**Figure 5 healthcare-09-00140-f005:**
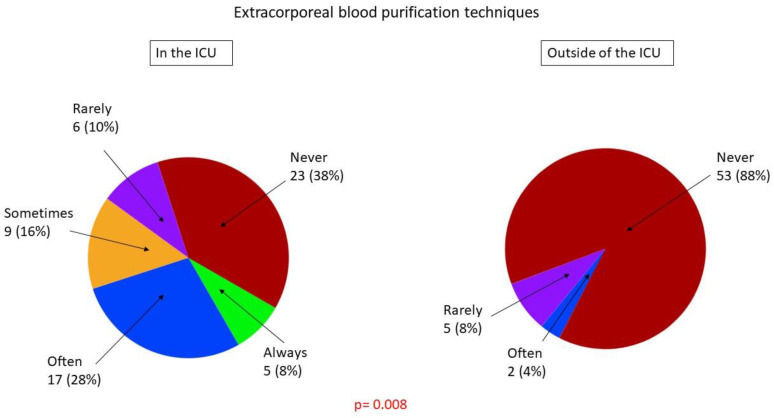
The use of extracorporeal blood purification techniques as an adjunct to hemodynamic support.

**Table 1 healthcare-09-00140-t001:** Basic data regarding hospitals participating in the study.

Variable		*n* (%)
No. of beds in the hospital	<100	6 (10%)
	100–250	16 (27%)
	251–500	23 (38%)
	>500	15 (25%)
No. of beds in the ICU	<6	21 (35%)
	6–10	25 (42%)
	>10	14 (23%)
Sepsis admissions to the ICU (interdepartmental transfers, within the hospital)	<10% of admissions	24 (40%)
	10–30% of admissions	21 (35%)
	31–50% of admissions	10 (17%)
	>50% of admissions	5 (8%)
Sepsis admissions to the ICU (external transfers)	<10% of admissions	48 (80%)
	10–30% of admissions	10 (16%)
	31–50% of admissions	1 (2%)
	>50% of admissions	1 (2%)
Rapid Response Teams available in the hospital	Yes	10 (17%)
	No	50 (83%)

**Table 2 healthcare-09-00140-t002:** Strategies regarding antimicrobial treatment in sepsis.

Procedure		Given Answer	In the ICU	Outside the ICU	*p*
Blood cultures taken when sepsis is suspected		Always	58 (96%)	29 (48%)	0.018
		Often	1 (2%)	21 (35%)	
		Sometimes	1 (2%)	6 (10%)	
		Rarely	0	4 (7%)	
		Never	0	0	
Hospital-based standard operating procedures for antimicrobial treatment		Yes	46 (77%)	36 (60%)	<0.001
		No	14 (23%)	24 (40%)	
Modification of antimicrobial treatment post-ICU admission	When transfer within the hospital is applied	Always	5 (8%)	N/A ^1^	-
		Often	37 (62%)		
		Sometimes	14 (24%)		
		Rarely	4 (6%)		
		Never	0		
	When an external transfer is applied	Always	8 (13%)	N/A	-
		Often	34 (57%)		
		Sometimes	16 (26%)		
		Rarely	2 (4%)		
		Never	0		
Reasons for modification of empiric therapy	Lack of therapeutic effects	Always	14 (23%)	N/A	-
		Often	38 (63%)		
		Sometimes	7 (12%)		
		Rarely	1 (2%)		
		Never	0		
	Epidemiological situation in the hospital/region	Always	10 (17%)	N/A	-
		Often	20 (33%)		
		Sometimes	14 (24%)		
		Rarely	12 (20%)		
		Never	4 (6%)		
	Contraindications/adverse effects	Always	13 (22%)	N/A	-
		Often	2 (4%)		
		Sometimes	16 (26%)		
		Rarely	24 (40%)		
		Never	5 (8%)		
	Wider therapeutic options available in the ICU	Always	13 (22%)	N/A	-
		Often	27 (45%)		
		Sometimes	9 (15%)		
		Rarely	7 (12%)		
		Never	4 (6%)		
Prebiotics/probiotics use		Always	14 (23%)	4 (7%)	<0.001
		Often	12 (20%)	14 (23%)	
		Sometimes	9 (15%)	16 (27%)	
		Rarely	5 (8%)	18 (30%)	
		Never	20 (34%)	8 (13%)	

^1^ N/A—not applicable.

**Table 3 healthcare-09-00140-t003:** Hemodynamic management in sepsis and septic shock—fluid responsiveness assessment and types of implemented fluids.

Procedure		Given Answer	In the ICU	Outside the ICU	*p*
Hospital-based fluid therapy algorithms		Yes	N/As ^1^	8 (13%)	-
		No		52 (87%)	
Fluid responsiveness assessment	Clinical state	Always	54 (90%)	36 (60%)	0.003
		Often	4 (6%)	17 (28%)	
		Sometimes	1 (2%)	5 (8%)	
		Rarely	1 (2%)	2 (4%)	
		Never	0	0	
	Arterial blood pressure	Always	54 (90%)	40 (67%)	0.011
		Often	4 (7%)	20 (33%)	
		Sometimes	2 (3%)	0	
		Rarely	0	0	
		Never	0	0	
	Diuresis	Always	54 (90%)	31 (52%)	0.48
		Often	4 (7%)	20 (33%)	
		Sometimes	0	8 (13%)	
		Rarely	2 (3%)	1 (2%)	
		Never	0	0	
	Lactate concentration	Always	43 (72%)	7 (12%)	0.32
		Often	11 (18%)	12 (20%)	
		Sometimes	4 (6%)	11 (18%)	
		Rarely	1 (2%)	22 (37%)	
		Never	1 (2%)	8 (13%)	
	Capillary refill time	Always	14 (23%)	4 (6%)	0.003
		Often	14 (23%)	7 (12%)	
		Sometimes	15 (25%)	8 (13%)	
		Rarely	11 (18%)	26 (44%)	
		Never	6 (11%)	15 (25%)	
		Often	7 (12%)	0	
		Sometimes	9 (15%)	1 (2%)	
		Rarely	7 (12%)	56 (93%)	
		Never	37 (61%)	3 (5%)	
Fluid therapy	Balanced crystalloids	Always	52 (87%)	24 (40%)	0.09
		Often	8 (13%)	30 (50%)	
		Sometimes	0	5 (8%)	
		Rarely	0	1 (2%)	
		Never	0	0	
	Unbalanced crystalloids	Always	3 (5%)	4 (6%)	0.018
		Often	1 (2%)	16 (27%)	
		Sometimes	9 (15%)	20 (33%)	
		Rarely	29 (48%)	18 (30%)	
		Never	18 (30%)	2 (4%)	
	Colloids (any)	Always	4 (6%)	2 (4%)	<0.001
		Often	11 (18%)	9 (15%)	
		Sometimes	8 (13%)	13 (21%)	
		Rarely	18 (30%)	22 (37%)	
		Never	19 (33%)	14 (23%)	
		Rarely	8 (13%)	21 (35%)	
		Never	14 (23%)	27 (45%)	

^1^ N/As—not assessed.

**Table 4 healthcare-09-00140-t004:** Hemodynamic management in sepsis and septic shock—use of catecholamines and vasopressors in septic shock treatment.

Procedure		Given Answer	In the ICU	Outside the ICU	*p*
Central venous catheterization before vasopressor infusion		Always	N/As ^1^	12 (20%)	-
		Often		24 (40%)	
		Sometimes		16 (26%)	
		Rarely		6 (10%)	
		Never		2 (4%)	
Use of catecholamines and vasopressors	Dopamine	Always	2 (4%)	4 (6%)	<0.001
		Often	7 (12%)	22 (37%)	
		Sometimes	13 (21%)	10 (17%)	
		Rarely	18 (30%)	18 (30%)	
		Never	20 (33%)	6 (10%)	
	Dobutamine	Always	4 (6%)	4 (6%)	<0.001
		Often	19 (32%)	17 (28%)	
		Sometimes	24 (40%)	13 (22%)	
		Rarely	9 (15%)	23 (39%)	
		Never	4 (6%)	3 (5%)	
	Norepinephrine	Always	51 (85%)	26 (43%)	0.02
		Often	9 (15%)	19 (32%)	
		Sometimes	0	7 (12%)	
		Rarely	0	5 (8%)	
		Never	0	3 (5%)	
	Epinephrine	Always	6 (10%)	2 (4%)	<0.001
		Often	15 (25%)	4 (6%)	
		Sometimes	25 (41%)	11 (18%)	
		Rarely	12 (20%)	21 (35%)	
		Never	2 (4%)	22 (37%)	
	Argipressin	Always	0	0	<0.001
		Often	7 (12%)	0	
		Sometimes	9 (15%)	1 (2%)	
		Rarely	7 (12%)	56 (93%)	
		Never	37 (61%)	3 (5%)	
	Terlipressin	Always	0	0	<0.001
		Often	5 (8%)	0	
		Sometimes	12 (20%)	4 (6%)	
		Rarely	16 (27%)	12 (20%)	
		Never	27 (45%)	44 (74%)	
		Rarely	18 (30%)	22 (37%)	
		Never	19 (33%)	14 (23%)	
		Often	24 (40%)	17 (28%)	
		Sometimes	16 (27%)	23 (38%)	
		Rarely	9 (15%)	14 (23%)	
		Never	0	5 (9%)	

^1^ N/As—not assessed.

## Data Availability

The data presented in this study are available on request from the corresponding author.

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
