# Peer review of "What Do We Know about Early Management of Sepsis and Septic Shock in Polish Hospitals? A Questionnaire Study"

_healthcare, 2021, doi:10.3390/healthcare9020140_

Round 1

Reviewer 1 Report

The authors conducted a survey study in the three largest regions (viovodeships) in Poland regarding knowledge and perceived compliance to the sepsis care bundles supported by the Surviving Sepsis Campaign. This survey assesses important items when considering structured quality improvements and identification of potential hurdles in implementation of sepsis care bundles. This is an important first step in a nationwide sepsis quality improvement and sepsis education project. 

The survey was sent to ICU chief medical officers (CMO), the better term may be ICU directors of the identified regions.  CMO usually describes the lead of a hospital, please make sure that this is described in the correct terms. 

 The word voivodeship will not be understood by readers unfamiliar with the Polish language and should be changed throughout the manuscript and potentially replaced by the word "region" for easier understanding of the reader.

The authors should consider moving the mentioning of the one hours sepsis bundle into the discussion. This concept has been strongly disputed in the United States as impractical to archive due to various logistical challenges. Dr Mitchell Levy is currently applying for funding for a structured investigation of this concept with help of the Society of Critical Care Medicine in US hospitals. The concept shortens the time to archive the elements of a 3 hour bundle: Initial Lactate, blood cultures prior to antibiotics, appropriate and broad spectrum antibiotics within 1 hours and if shock is present by hypotension or abnormal lactate measurements of greater than 4 mg/ dL the administration of 30 cc per kg bodyweight. Especially the application of the crystalloid fluid bolus requirement has been challenged in the setting of the Emergency Department. This is an important item to mention, however would be better in the discussion with additional review of the literature in regard to early goal directed therapy [ Rivers EGDT trial NEJM 2001, ProCESS, ARISE and ProMISe trials, FACTT trial]. 

The presentation of the results should consider graphics and tables for a better visual and a better structured presentation of the collected data. Consider Pie and Bar graphs when possible and also consider moving some of the tables to a supplemental file. Additionally, consider showing your surveyed regions on a map as a heat map for the population density.

Line 113 mentions NEWS2. Please give the appropriate reference for this and a brief description of what this entails. The same should be done for qSOFA. It is not clear what the inflammatory markers entail: C Reactive Protein, Sedimentation Rate (ESR), procalcitonin or various cytokines such as Interleukin 6, 8 and 1 or Tumor Necrosis factor. A brief description on what that term means will help to put this in a better context.

Antimicrobial treatment in sepsis has various aspects. The initial treatment should be broad enough to cover the most prevalent organisms for the septic episode during the initial resuscitation. Following the culture results de-escalation and narrowing of therapeutics should be considered. The question regarding therapeutic effects appears more related to the the later phases of source control and not the initial resuscitation. This may be an important item to consider in the discussion. 

Table 3 contains the term pre-biotics and probiotic use. This term should be explained some more and should be supported by references.

Table 4 is very long. This needs to be presented differently as this is difficult for the reader to follow. Graphic summaries may be helpful.

That there were no hospital-based fluid therapy algorithms in most hospitals needs also more discussion. 

For your discussion consider describing the most important elements of Sepsis Resuscitation and Sepsis Maintenance bundles you surveyed.

Consider including care bundle successes including Leishman New York Hospital Association data published in the Journal of Critical Care Medicine.

                Consider rephrasing the sentence starting on line 182: Incompliance = Non-Compliance. The issues with the reduced compliance are mainly related to delayed recognition which leads to delays in treatment initiation and therefore missing treatment timeframes/ delays. Most sepsis care bundles are all or none and with delays are difficult to archive.

A table with the elements of sepsis care bundles will help here to carry the points

                3 HOUR Bundle

  1. Early Recognition (SIRS criteria, qSOFA, NEWS2, electronic medical records screening)
  2. Risk Stratification with Blood pressure and Lactate/ Lactic Acid
  3. Screening for a source of infection including blood cultures and other appropriate cultures prior to the administration of antibiotics
  4. Appropriate antibiotics (board spectrum, source and local antibiotic resistance specific.)
  5. If the patient is identified as potential shock give fluids at 30 cc/ kg within 3 hours and re-assess with
  6. Lactate and Blood pressure assessments

6 HOUR Bundle

  1. If the shock state should continue consider administration of vasoactive medication
  2. Consider more advanced shock and resuscitation assessments
    1. Central Line – CVP and ScVO2
    2. Dynamic Fluid Assessments
    3. Physical Exam
    4. Ultrasound Assessments
  3. Administration of vasoactive agents to improve perfusion blood pressure

MAINTANANCE BUNDLES

  1. Steroids, Thiamine, Vitamin C
  2. Extracorporeal blood purification
  3. Prebiotics/ Probiotics
  4. Other Care Bundles for mechanical Ventilation, Sedation, early mobilization

Author Response

Reviewer #1

We would like to thank you for your extensive review and valuable input in improving our study. We are glad you see the importance of the conducted study in the improvement of nationwide sepsis management. Please find below our responses to your comments and corrections.

  1. We modified the term according to your suggestion in favour of the „ICU director” in place of the „ICU chief medical oficer”.
  2. We removed the word „voivodeship” and exchanged it to „region”, as suggested.
  3. We further expanded upon the issue of sepsis care bundles in the discussion, after mentioning it in the introduction, and additional supplementary material was added in form of the Table S2. We elaborated on the ongoing issue of fluid therapy application based on the suggested source clinical trials through a brief summary regarding their key findings and impact on the development of sepsis care bundles.
  4. We followed your suggestions and diversified the way we present our results. We modified the existing tables in favour of adding pie charts and bar graphs, we reduced the numer of tables in the main manuscript text by moving some of the more complex tables to the supplementary material. We presented our surveyed regions on a figure representing their location within Polish borders.
  5. We expanded upon the meaning and construction of utilized scores. We described with more details the meaning of the inflammatory markers with clarification regarding which markers were considered and analyzed most frequently. Please find the extension of this issue in the Figure 2 description and the manuscript text.
  6. Thank you for stressing the importance of the issue of antimicrobial treatment. We included your valuable input and comment in the discussion section of our manuscript.
  7. We included a brief definition of the terms prebiotics and probiotics, supported by references, as suggested.
  8. We reduced the amount of data shown as a table, and transferred parts of them to form smaller tables and graphs.
  9. We added further arguments explaining the lack of hospital-based fluid therapy algorithms in the discussion section of our paper.
  10. As mentioned in the previous response to your comments, we decided to include the detailed elements of sepsis care bundles in the supplementary material.
  11. We rephrased the sentence according to your suggestions.

With regards,

the Authors

Reviewer 2 Report

This is a clearly presented study describing the situation with sepsis diagnosis and treatment in three regions in Poland.

The study presents an extensive study and is clearly written. The comparison between different regions of Poland is probably more interesting for Polish health givers than others. 

The Tables are clear but maybe too detailed and as suggested to the authors, the manuscript would be easier to read if the data would be presented in a simplified way - eg as Figures. 

There are extensive detailed Tables and maybe they could be reduced in the following way without too much loss of information: always and often could be merged to one section, sometimes remain as it is and rarely and never merged to one section. This would decrease the number of lines per item from 5 to 3.

Another way to simplify the presentation would be to use Figures with "pies" (donuts) with different sections for always, often, sometimes, rarely and never, respectively and use, for instance,  the inner section of the pie for ICU and the outer for outside the ICU - thus visualising the data in a clear way.

Some of the abbreviations are not explained when they appear for the first time: line 135: VICTAS, line 192 qSOFA score.

Author Response

Reviewer #2

We would like to thank you for your review, comments and observations. We followed through with your suggestions. Please, find below the summary of implemented revisions and corrections.

  1. According to your suggestions, we verified the way we showed our results, and diversified the means of depicting them. We modified tables, moved some of the more extensive tables to the supplementary material. We utilized pie charts and bar graphs to improve the reception of the presented results and data.
  2. We explained the abbreviations.

With regards,

the Authors